# Fecal Calprotectin and Segmental Inflammation in Small Bowel Crohn’s Disease: A Capsule Endoscopy-Based Study

**DOI:** 10.3390/biomedicines13092179

**Published:** 2025-09-06

**Authors:** Mariana Souto, Ana Isabel Ferreira, João Gonçalves, Cátia Arieira, Francisca Dias de Castro, Vitor Macedo Silva, Bruno Rosa, José Cotter

**Affiliations:** 1Gastroenterology Department, Unidade Local de Saúde do Alto Ave, 4835-044 Guimarães, Portugal; anaisabelferreira@ulsaave.min-saude.pt (A.I.F.); joaogoncalves@ulsaave.min-saude.pt (J.G.); catiaarieira@ulsaave.min-saude.pt (C.A.); franciscacastro@ulsaave.min-saude.pt (F.D.d.C.); vitormacedo@ulsaave.min-saude.pt (V.M.S.); brunorosa@ulsaave.min-saude.pt (B.R.); jabcotter@gmail.com (J.C.); 2Life and Health Sciences Research Institute (ICVS), School of Medicine, University of Minho, 4710-057 Braga, Portugal; 3ICVS/3B’s–PT Government Associate Laboratory, 4710-057 Braga, Portugal

**Keywords:** Crohn’s disease, fecal calprotectin, capsule endoscopy, Lewis Score, small bowel inflammation

## Abstract

**Background/Objectives**: The correlation between fecal calprotectin (FC) levels and small bowel (SB) inflammation in Crohn’s Disease (CD) remains a subject of debate. This study aims to investigate the association between FC and SB inflammatory activity. **Methods**: Retrospective cohort study involving patients with SB-CD who underwent small bowel capsule endoscopy (SBCE) and FC testing, excluding those with colonic inflammation. Patients were categorized based on the Lewis Score (LS): no inflammation (all SB tertiles with LS < 135); proximal SB inflammation (first and/or second tertiles with LS ≥ 135, without inflammation in the third tertile); distal SB inflammation (third SB tertile with LS ≥ 135, no inflammation in the proximal SB); or pan-SB inflammation (proximal and distal SB with inflammation). **Results**: Eighty-seven patients were included (median age 35 years, 75.9% female). Inflammation was absent in 21.8% of patients, proximal inflammation in 4.6%, distal inflammation in 33.3% and pan-SB inflammation in 40.4%. FC median values exhibited an ascending trend along the SB: no inflammation 58 µg/g; proximal SB inflammation 65 µg/g; distal SB inflammation 122 µg/g; or pan-SB inflammation 400 µg/g. FC correlated with LS in the second (ρ = 0.464) and third tertiles (ρ = 0.435), but not in the first tertile. FC levels were significantly higher in pan-SB disease compared to isolated proximal (*p* = 0.014) and distal inflammation (*p* = 0.012). A cutoff of 178 µg/g differentiated pan-SB from isolated distal disease (AUC = 0.716; sensitivity 85.7%, specificity 58.6%). **Conclusions**: FC levels correlate positively with the presence of SB lesions in the second and third tertiles. However, it is not a reliable marker for detecting inflammation in the first tertile, highlighting the importance of performing a SBCE in these patients.

## 1. Introduction

Crohn’s disease (CD) is a chronic, relapsing inflammatory condition that can affect any segment of the gastrointestinal tract. In particular, the small bowel (SB) is involved in up to 80% of cases, and approximately 30% of patients present with inflammation confined exclusively to the SB [1]. Effective diagnosis and disease monitoring require a combination of invasive and noninvasive tools, including cross-sectional imaging, endoscopy, and inflammatory biomarkers [2,3].

Fecal calprotectin (FC) is one of the most widely used noninvasive biomarkers in inflammatory bowel disease (IBD). It is recommended by both the European Crohn’s and Colitis Organization (ECCO) and the American College of Gastroenterology’s (ACG) guidelines for differentiating IBD from Irritable Bowel Syndrome, assessing mucosal healing, monitoring treatment response, and predicting relapse or postoperative recurrence [2,3,4]. FC reflects neutrophil activity and intestinal inflammation by quantifying calprotectin, a calcium-binding protein released by activated neutrophils during mucosal injury [5].

While FC has demonstrated high diagnostic accuracy for colonic CD and Ulcerative Colitis, its role in small bowel Crohn’s disease (SB-CD) remains debated. Several studies have reported a positive correlation between FC levels and endoscopic disease activity in SB-CD [6,7,8,9], but others have found limited sensitivity and specificity, particularly in cases with either ileal or proximal small bowel involvement [10,11]. This inconsistency may, in part, be explained by the majority of studies evaluating FC in the context of colonic inflammation or treating the small bowel as a single anatomical unit without addressing segmental variability.

Notably, disease location within the SB may have significant clinical implications. Proximal CD, often affecting the jejunum or duodenum, is associated with a more aggressive disease course, higher complication rates, and an increased need for surgery [12,13,14]. In contrast, distal SB involvement tends to follow a more indolent progression. Moreover, jejunal disease may be underdiagnosed, given its anatomical inaccessibility to conventional endoscopic techniques [15].

Despite its clinical relevance, only one study to date has specifically explored FC performance along different SB segments [16], suggesting that location-specific variation in FC levels could significantly impact its clinical utility.

Given these gaps, the present study aims to investigate the association between FC levels and the topographical distribution of SB inflammation, as evaluated by a small bowel capsule endoscopy (SBCE). By focusing on segment-specific disease involvement, this work seeks to offer more precise insights into the utility of FC in monitoring and managing SB-CD.

## 2. Materials and Methods

### 2.1. Study Design and Population

A retrospective, single-center study was conducted, including adult patients with previously diagnosed CD, based on clinical, radiological, endoscopic, and histopathological criteria [3]. SBCE was performed as part of disease monitoring, and only patients with a quantified Lewis Score (LS) were included [17,18]. The study period spanned between January 2013 and December 2021. Patients were included if they had L1 disease (with or without L4) according to the Montreal classification [19] and had a FC measurement within three months of the SBCE procedure, with no change in therapy during this period. Patients with L2 and L3 (colonic or ileocolic) disease or stricturing/fistulizing phenotypes at presentation were excluded. SBCE examinations were considered incomplete if the capsule failed to reach the cecum within the allotted recording time, such cases were excluded. Demographic, clinical, endoscopic, and laboratory data were retrieved through a review of medical records.

Fecal calprotectin was measured using the Calprotectin Quantum Blue^®^ assay (Bühlmann, Schönenbuch, Switzerland), a quantitative immunochromatographic method with automated reading. The same assay and protocol were applied consistently throughout the study period.

A diagram summarizing the study sample selection is presented in Figure 1.

### 2.2. SBCE Procedure

SBCE was performed using the PillCam^®^ SB2 system (January 2013–December 2013) or the PillCam^®^ SB3 system (January 2014–December 2021) (Given Imaging—Medtronic, Yokneam, Israel). Patients with a high risk of capsule retention—such as those presenting with obstructive symptoms, a history of small bowel resection, or known strictures—were excluded. All enrolled patients were instructed to follow a clear liquid diet for 24 h before SBCE and to fast for 12 h before capsule ingestion. After ingesting the capsule, patients were permitted to consume clear liquids at two hours and a light meal at four hours, contingent upon confirmation of the capsule’s passage into the SB via the Real Time Viewer^®^ (Given Imaging—Medtronic). Oral iron supplementation was discontinued for seven days and nonsteroidal anti-inflammatory drugs for one month before undergoing SBCE.

### 2.3. Assessment of Mucosal Inflammation Small Bowel Activity

Each video capsule endoscopy was interpreted by two experienced readers. The SB was divided into three tertiles—first, second, and third—based on capsule transit time determined by the Rapid Reader software v9. For each tertile, the segmental LS was calculated according to standard criteria, including villous appearance (e.g., edematous vs. normal), the number and extent of mucosal ulcers, and the presence of strictures. Inflammatory activity was categorized as follows:-LS < 135: no inflammation or clinically insignificant inflammation.-LS ≥ 135: presence of mucosal inflammation.

### 2.4. Criteria and Definitions of Disease Anatomical Involvement

In accordance with the Montreal L4 classification [19], proximal CD—which is often linked to a worse prognosis—was defined as involvement of the first and/or second small bowel tertiles, labeled as “Proximal SB.” Involvement of the third tertile was classified as “Distal SB.”

Based on anatomical extent, patients were categorized into four subgroups:No inflammation: LS < 135 in all three SB tertiles.Proximal SB inflammation: LS ≥ 135 in the first and/or second SB tertiles, with no inflammation in the third SB tertile (LS < 135).Distal SB inflammation: LS ≥ 135 in the third SB tertile, with no inflammation in the first and/or second SB tertiles (LS < 135).Pan-SB inflammation: LS ≥ 135 in both the first and/or second plus third SB tertiles.

These categories were used to compare median FC levels.

### 2.5. Statistical Analyses

Statistical analysis was performed using IBM^®^ SPSS^®^ Statistics 26.0 software (Armonk, NY, USA).

Categorical variables were presented as frequencies and proportions (%). Continuous variables were first tested for normality using the Shapiro–Wilk test. Non-parametric continuous data were presented as a median and interquartile range (IQR). Differences in FC levels across the four inflammation groups (no inflammation, proximal, distal, and pan-SB inflammation) were assessed using the Kruskal–Wallis test, with post hoc pairwise comparisons between groups using the Bonferroni-adjusted Mann–Whitney U test. The same approach was applied to compare median LS values across inflammation groups. Given the ordinal nature of the variable extent of inflammation (no inflammation < distal inflammation < pan-SB inflammation), we used the Jonckheere–Terpstra test to evaluate for a monotonic trend in FC values. Patients with isolated proximal disease (*n* = 4) were excluded due to small sample size.

To explore the relationship between FC levels and ongoing therapy, group comparisons were performed using the Kruskal–Wallis test.

Correlations between each of the tertile’s LS to FC levels were obtained using Spearman’s correlation. Sensitivity analyses were conducted by restricting correlation analyses to patients with FC collected within ≤30 and ≤60 days of SBCE. Additionally, to evaluate whether the association between FC and SB inflammatory activity was independent of potential confounders, a multivariable linear regression model was constructed using a log-transformed FC as the dependent variable. The independent variable of interest was the LS, and covariates included therapy, C-reactive protein (CRP), smoking status, perianal disease, age, and sex.

A receiver operating characteristic (ROC) curve was constructed, and the area under the curve (AUC) was calculated to explore the discriminatory accuracy of FC in different anatomical extent disease activity. Youden’s most accurate points were computed for each ROC curve, as well as sensitivity and specificity. All statistical tests were two-sided, and *p* = 0.05 was considered statistically significant.

## 3. Results

### 3.1. Patients’ Characteristics

A total of 87 patients were included in the study, with a median age of 35 years (IQR: 24–46). The majority were female (*n* = 66; 75.9%). Smoking history was reported by 14 (16.1%) patients. Perianal disease was present in 10 (11.5%), and 25 (28.7%) had extraintestinal manifestations. Regarding treatment, 35 patients (40.2%) were not receiving active therapy, 17 (19.5%) were on biologic therapy, 20 (23.0%) on immunomodulators, 8 (9.2%) on topical corticosteroids, 3 (3.4%) on systemic corticosteroids, and 4 (4.6%) were receiving combined therapy with biologics and immunomodulators.

Based on capsule endoscopy findings, 19 (21.8%) patients had no detectable inflammation, 4 (4.6%) had isolated proximal inflammation, 29 (33.3%) had isolated distal inflammation, and 35 (40.2%) had pan-small bowel inflammation. The overall LS median was 337 (IQR: 135–900). When stratified by tertiles, the first tertile had a median score of 0 (IQR: 0–135), the second tertile 0 (IQR: 0–225), and the third tertile 337 (IQR: 20–900).

Median FC was 205 µg/g (IQR: 58–506).

Full patient characteristics are summarized in Table 1.

The median interval between FC sampling and SBCE was 51 days (IQR 23–90; minimum 0, maximum 90). Overall, 33 patients (37.9%) had FC collected within 30 days of SBCE, and 56 (64.4%) within 60 days.

### 3.2. Fecal Calprotectin Levels and Distribution of Inflammation

Median FC levels varied according to the location and extent of SB inflammation. Patients with no detectable inflammation had a median FC level of 58 µg/g (IQR: 33–180; minimum 30, maximum 261), while those with isolated proximal inflammation had a median of 65 µg/g (IQR: 30–128; minimum 30, maximum 137). Patients with distal SB inflammation showed higher FC levels with a median of 122 µg/g (IQR: 42–507; minimum 30, maximum 1800), and those with pan-SB involvement had markedly elevated levels, with a median of 400 µg/g (IQR: 205–852; minimum 42, maximum 1800). These results are represented in Figure 2A.

Median FC levels were statistically significantly higher in patients with more extensive inflammation (pan-SB involvement) compared to those with isolated distal (*p* = 0.012) or proximal small bowel inflammation (*p* = 0.014), despite no significant differences observed in the corresponding LS (*p* = 0.329 and *p* = 0.084, respectively), as observed in Figure 2B.

Although patients with distal inflammation had numerically higher FC levels compared to those with no inflammation or isolated proximal inflammation, these differences were not statistically significant (*p* = 0.456 and *p* = 0.714, respectively). Similarly, no significant difference was observed between patients without inflammation and those with isolated proximal inflammation (*p* = 1.000).

The Jonckheere–Terpstra test confirmed a significant positive trend in FC values with increasing extent of inflammation (no inflammation < distal < pan-SB; T_JT_ = 1675, z = 4.73, *p* < 0.001).

Additionally, we explored FC levels according to ongoing therapy (no therapy, corticosteroids, immunomodulators, biologics, or combined therapy). No significant differences in median FC values were observed across treatment groups (*p* = 0.316).

### 3.3. Correlation Between Lewis Score and Fecal Calprotectin

In multivariable linear regression models, the LS remained independently associated with FC levels (*p* = 0.036), after adjustment for potential confounders. Among the potential confounders, only CRP showed an independent association (*p* = 0.007).

To further explore this association by bowel segment, correlation analysis between LS and FC levels across SB tertiles revealed no significant correlation in the first tertile (*p* = 0.479). However, moderate and statistically significant correlations were observed in both the second (ρ = 0.464, *p* < 0.001) and third tertiles (ρ = 0.435, *p* < 0.001).

In sensitivity analyses restricted to patients with shorter intervals between FC and SBCE, the overall findings were consistent. Among patients tested within ≤30 days, no significant correlation was observed between FC and LS in the first tertile (*p* = 0.177), whereas significant correlations were confirmed in the third tertile (ρ = 0.490, *p* = 0.004). Similarly, within ≤60 days, FC remained significantly correlated with LS in the second (ρ = 0.363, *p* = 0.006) and third tertiles (ρ = 0.407, *p* = 0.002), but not in the first tertile (*p* = 0.248).

Representative SBCE images of inflammatory lesions in the first, second, and third small bowel tertiles are shown in Figure 3.

### 3.4. Fecal Calprotectin Diagnostic Yield

When comparing patients with any inflammation versus no inflammation, FC showed a fair discriminative ability, with an AUC of 0.753 (95% CI: 0.649–0.840, *p* < 0.001). After excluding patients with isolated proximal inflammation, the diagnostic performance improved, with an AUC of 0.780 (95% CI: 0.676–0.864, *p* < 0.001). At the optimal cutoff of >228 µg/g, sensitivity was 57.8% and specificity was 94.7%, with a positive predictive value (PPV) of 97.4% and a negative predictive value (NPV) of 40.0%.

An FC cutoff of 178 µg/g was identified as the optimal threshold to distinguish patients with pan-SB inflammation from those with isolated distal SB inflammation, with an AUC of 0.716 (95% CI: 0.586–0.846, *p* = 0.001), sensitivity of 85.7% and specificity of 58.6%, PPV of 71.4% and NPV of 77.3%.

Exploratory subgroup analyses confirmed the limited diagnostic utility of FC in isolated proximal SB disease, with an AUC of 0.671 (95% CI: 0.446–0.850, *p* = 0.248). Conversely, in patients with distal SB inflammation versus no/proximal inflammation, FC achieved an AUC of 0.654 (95% CI: 0.509–0.780, *p* = 0.043).

## 4. Discussion

This study aimed to assess the diagnostic utility of FC in detecting segment-specific SB disease activity in patients with CD using SBCE as a standard reference. Our findings revealed that FC levels progressively increased with the extent of inflammation, with the highest concentrations observed in patients with pan-SB involvement. These results reinforce previous reports indicating that FC correlates with the extent and severity of mucosal inflammation in CD [6,8].

Our data further demonstrated a clear positive trend of FC levels according to the extent of small bowel inflammation, confirmed by the Jonckheere–Terpstra test (*p* < 0.001). This ascending gradient, from no inflammation to isolated distal and pan-SB disease, reinforces the biological plausibility that FC reflects an increasing inflammatory burden. Nevertheless, our data also highlight a critical limitation of FC: its limited performance in detecting isolated proximal SB inflammation in this cohort. Median FC levels in patients with proximal disease were not significantly different from those without inflammation, and no correlation was observed between FC and LS in the first small bowel tertile. This observation aligns with findings from previous studies, who reported that FC correlates well with active colonic inflammation but not with proximal CD activity [6,10,11,20].

Several mechanisms were proposed by Zittan et al., who noted that patients with isolated SB-CD often exhibit shorter segments of inflammation and reduced neutrophil shedding, potentially leading to lower FC levels despite active disease [10]. Additionally, differences in motility and transit time—faster in the SB, when compared to the colon—may further reduce calprotectin stability. This phenomenon underscores the anatomical and physiological limitations of FC as a biomarker and supports the need for complementary diagnostic tools, particularly in patients with suspected proximal CD and non-elevated FC levels.

The small number of patients with isolated proximal CD in our cohort (4.6%) reflects the rarity of this phenotype. A large study from the IBD Genetics Consortium, which included 2105 patients with confirmed CD, reported that only 25 individuals (1.2%) had isolated proximal disease without ileal or colonic involvement [12]. Thus, low numbers are expected even in large series. Accordingly, our results in this subgroup should be interpreted as exploratory and hypothesis-generating: the limited sample size reduces statistical power and increases the instability of estimates.

Importantly, the only study to date that has examined the association between FC levels and segmental localization of inflammation within the SB is the recent work by Ukashi et al. [16]. In a post hoc aggregated analysis of five prospective studies, the authors confirmed a diagnostic level gradient of FC along the SB, showing that FC has higher sensitivity for distal inflammation compared to proximal disease. Their work represents a valuable contribution to the field and reinforces the importance for a more refined interpretation of FC in clinical practice. Nonetheless, as their analysis included a heterogeneous CD population—with frequent colonic involvement—and was conducted post hoc across multiple study protocols, some limitations regarding patient selection and methodological consistency may be considered. In contrast, our study offers a novel perspective by focusing specifically on a well-defined cohort of patients with CD limited exclusively to the SB, thereby providing a more targeted assessment of FC’s diagnostic performance in this anatomically restricted phenotype.

Moreover, the optimal FC cutoffs for detecting SB inflammation remain highly variable across studies. Previous meta-analyses have suggested thresholds ranging from 76 to 265 µg/g for small bowel disease detection, depending on methodology and population characteristics [21,22]. In our cohort, a cutoff of >228 µg/g yielded a fair diagnostic performance for detecting any inflammation versus none, in line with these ranges (AUC of 0.753, *p* < 0.001). Notably, diagnostic performance improved when patients with isolated proximal disease were excluded, with the AUC rising to 0.780, suggesting that FC may be less informative for proximal inflammation. This interpretation was further supported by exploratory analyses: FC demonstrated a significant discriminative ability for distal SB inflammation (AUC 0.654, *p* = 0.043), whereas diagnostic performance for isolated proximal disease did not reach statistical significance (*p* = 0.248).

In addition, we identified a cutoff of 178 µg/g as the optimal threshold to differentiate patients with pan-SB inflammation from those with isolated distal disease, with high sensitivity (85.7%) and moderate specificity. This finding suggests that FC values above 178 µg/g may help clinicians identify patients at risk of more extensive disease, even when only limited distal inflammation is observed during ileocolonoscopy—thus, prompting further investigation with SBCE. These results highlight the value of individualized FC interpretation and support the use of segment-specific thresholds adapted to distinct clinical scenarios.

These nuanced findings highlight the limitations of a “one-size-fits-all” approach to FC interpretation. FC has been endorsed by both ECCO and ACG guidelines as a reliable surrogate marker for monitoring disease activity in IBD [2,3] and was even reported to be a reliable marker for mucosal healing in SB-CD [23]. However, the present results stress the importance of integrating FC with direct endoscopic assessments like SBCE—particularly given the increasing recognition of mucosal healing as a key therapeutic target in CD [4]. Recent evidence supports the role of SBCE not only in diagnosis, but also in disease monitoring. Ben-Horin et al. demonstrated that the inflammatory burden on SBCE predicts both short- and long-term flare risk [24], and that a SBCE-guided treat-to-target approach leads to improved outcomes compared to standard care [25].

Furthermore, the anatomical disease location has clinical implications. Proximal CD has been associated with a more aggressive phenotype and increased surgical risk [12,13,14]. Failure to detect proximal inflammation using FC alone could delay necessary interventions. Therefore, in patients with negative or borderline FC levels and ongoing symptoms, SBCE should be considered to exclude proximal disease [26].

Our study has several strengths, including the use of SBCE as the reference method, which allows for a high-resolution, segment-specific evaluation of mucosal inflammation throughout the SB. Notably, this is the first study to focus exclusively on patients with CD confined to the SB, offering a unique and targeted insight into the diagnostic performance of FC in this specific phenotype. This anatomically restricted approach enhances the clinical relevance and specificity of our findings. Nonetheless, some limitations should be acknowledged, including the retrospective design and the single-center setting, which may affect generalizability. Additionally, although all patients had FC measured within a three-month window of the SBCE, some degree of temporal variability between assessments may have influenced the results. The small number of patients with isolated proximal disease is an inherent limitation, although this is consistent with the rarity of this phenotype. Larger studies will be required to confirm these preliminary observations.

In conclusion, FC is a valuable, noninvasive biomarker for assessing mucosal inflammation in Crohn’s disease, particularly in cases of distal or extensive small bowel involvement. However, in this cohort, FC did not correlate with proximal inflammation, and further studies are warranted to clarify its diagnostic performance in this subgroup. These findings advocate for a stratified diagnostic approach that integrates FC with SBCE.

## Figures and Tables

**Figure 1 biomedicines-13-02179-f001:**
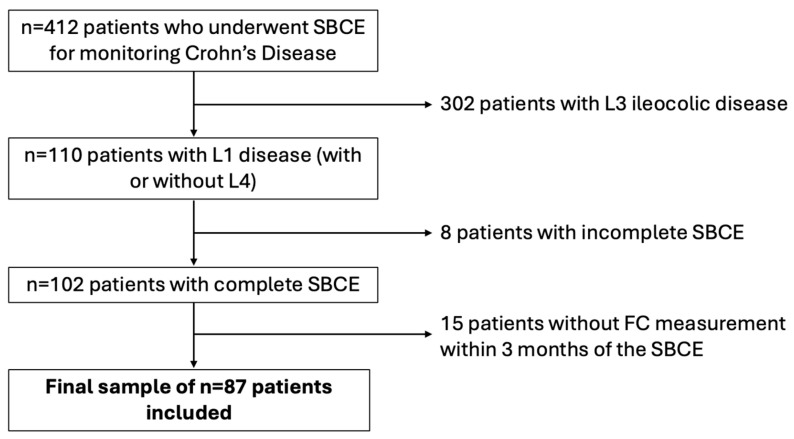
Diagram displaying patients’ selection process. SBCE—small bowel capsule endoscopy; FC—fecal calprotectin.

**Figure 2 biomedicines-13-02179-f002:**
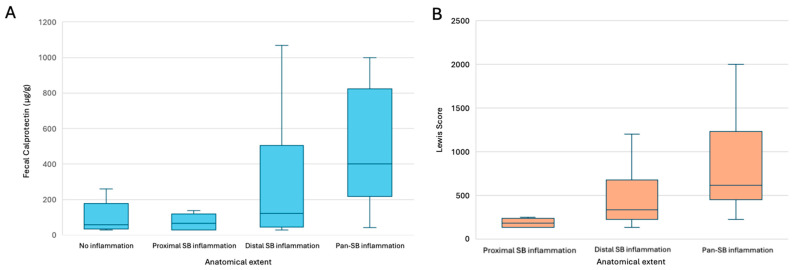
Median fecal calprotectin (**A**) and Lewis Score (**B**) across different anatomical extents of small bowel inflammation in Crohn’s disease. SB—small bowel.

**Figure 3 biomedicines-13-02179-f003:**
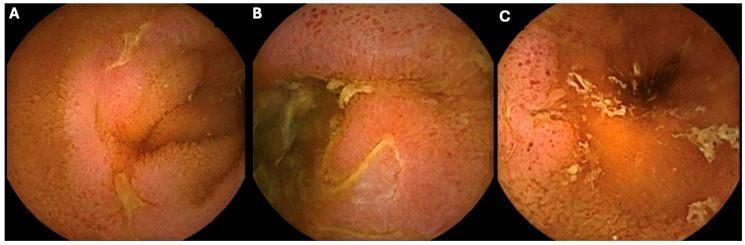
Representative capsule endoscopy findings in small bowel Crohn’s disease. (**A**) Inflammation in the first small bowel tertile; (**B**) inflammation in the second tertile; (**C**) inflammation in the third tertile.

**Table 1 biomedicines-13-02179-t001:** Patients’ Characteristic.

*	Total Cohort (*n* = 87)
Age (years)	35 (24–46)
Sex	
- Female	66 (75.9%)
Smoking history	14 (16.1%)
Age at diagnosis ^+^	
- Montreal A1	5 (5.7%)
- Montreal A2	61 (70.1%)
- Montreal A3	21 (24.1%)
Perianal involvemen	10 (11.5%)
Extraintestinal involvement	25 (28.7%)
Therapy	
- No therapy	35 (40.2%)
- Biologics	17 (19.5%)
- Immunomodulators	20 (23.0%)
- Topical corticosteroids ^⊥^	8 (9.2%)
- Systemic corticosteroids	3 (3.4%)
- Combined therapy (biologics and immunomodulators)	4 (4.6%)
Inflammation distribution	
- No inflammation	19 (21.8%)
- Pan-small bowel inflammation	35 (40.2%)
- Proximal inflammation	4 (4.6%)
- Distal inflammation	29 (33.3%)
Fecal calprotectin (µg/g)	205 (58–506)
Lewis Score	
- 1st tertile	0 (0–135)
- 2nd tertile	0 (0–225)
- 3rd tertile	337 (20–900)
- Overall	337 (135–900)

* Continuous variables are presented as median, with interquartile range (IQR) in parentheses; categorical variables are presented as absolute numbers, with percentages in parentheses. Minimum and maximum values for continuous variables were as follows: age 18–77 years, fecal calprotectin 30–1800 µg/g, and Lewis score by tertile: 1st tertile 0–458, 2nd tertile 0–1690, 3rd tertile 0–2140, overall 0–2140. ^+^ Age at diagnosis refers to Montreal classification: A1 (<17 years), A2 (17–40 years), A3 (>40 years). ^⊥^ Topical corticosteroids refer to oral budesonide formulations.

## Data Availability

The original contributions presented in this study are included in the article. Further inquiries can be directed to the corresponding author.

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
