# Peer review of "Fecal Calprotectin and Segmental Inflammation in Small Bowel Crohn’s Disease: A Capsule Endoscopy-Based Study"

_biomedicines, 2025, doi:10.3390/biomedicines13092179_

Round 1

Reviewer 1 Report

Comments and Suggestions for Authors

This is a well conceived and reported retrospective study comparing capsule endoscopy outcomes in Crohn's disease to faecal calprotectin results. The english language is accurate and very readable. The clinical question is important and, while there are multiple studies which have previously addressed the question, none have approached it in the methodological fashion taken here. However, there are a few points of clarification needed. 

On line 122 I am presuming what is meant here is that non-parametric continuous data, as defined by Shapiro-Wilk test, was presented as median and IQR. Currently the sentence seems to read the other way around.

On line 136, the results would benefit from a patient flow diagram (STROBE recommends reporting numbers at each stage). A diagram showing how many capsule procedures were considered, and how many were excluded with reasons, would strengthen transparency and help readers assess applicability.

Regarding table 1. table needs a footnote stating that the figures in brackets are the IQRs

Regarding Figure 1. It would be useful to see the distribution of Lewis scores alongside this. The median and IQRs are in the text but the ranges are not. I think the real problem here is that there are so few patients with isolated proximal disease (as would be expected) and those that do have so little activity that it really is difficult to draw any conclusions about the performance of FC for proximal disease which affects the applicability of the conclusions drawn. Another approach would be to make the raw FC and Lewis score data available as supplemental information so those who wished could get a real sense of the distribution of activity of the proximal disease.

At line 253  The reality is that comparing a group of 4 patients with isolated proximal disease to as much as 35 with pan small bowel disease has real problems

Low statistical power: With only 4 patients, the ability to detect true differences is minimal. You might miss real effects (Type II error).

Unstable estimates: Medians and IQRs from 4 people are not reliable representations of the “population.”

Multiple comparisons penalty: Bonferroni correction makes it even harder to detect significance, so results are even less stable with such a small group.

Risk of spurious results: A single outlier among 4 patients could drive the group median and make differences look larger than they really are.

The authors need to make this very clear to the reader and discuss how this affects the conclusion that FC is less reliable in proximal disease.

Author Response

Comment 1: On line 122 I am presuming what is meant here is that non-parametric continuous data, as defined by Shapiro-Wilk test, was presented as median and IQR. Currently the sentence seems to read the other way around.

Response 1: We thank the reviewer for this valuable observation. We agree that the original phrasing was ambiguous. The sentence has now been revised for clarity as follows (lines 130-132):

"Categorical variables were presented as frequencies and proportions (%). Continuous variables were first tested for normality using the Shapiro–Wilk test. Non-parametric continuous data were presented as median and interquartile range (IQR)."

This revised wording better reflects the statistical approach used and avoids misinterpretation.

Comment 2: On line 136, the results would benefit from a patient flow diagram (STROBE recommends reporting numbers at each stage). A diagram showing how many capsule procedures were considered, and how many were excluded with reasons, would strengthen transparency and help readers assess applicability.

Response 2: We thank the reviewer for this valuable suggestion. In accordance with the STROBE recommendations, we have now added a patient flow diagram (new Figure 1 – page 3) that details the total number of capsule endoscopy procedures performed for Crohn’s disease monitoring during the study period (n=412), the exclusions respective with reasons, and the final analyzed cohort (n=87). We believe this addition strengthens the transparency of our study and improves the clarity and applicability of the results.

Comment 3: Regarding table 1. table needs a footnote stating that the figures in brackets are the IQRs

Response 3: We appreciate the reviewer’s suggestion. We have added a footnote to Table 1 clarifying the format of the data: Continuous variables are presented as median (interquartile range, IQR) and categorical variables as absolute numbers (percentages).

Comment 4: Regarding Figure 1. It would be useful to see the distribution of Lewis scores alongside this. The median and IQRs are in the text but the ranges are not. I think the real problem here is that there are so few patients with isolated proximal disease (as would be expected) and those that do have so little activity that it really is difficult to draw any conclusions about the performance of FC for proximal disease which affects the applicability of the conclusions drawn. Another approach would be to make the raw FC and Lewis score data available as supplemental information so those who wished could get a real sense of the distribution of activity of the proximal disease.

Response 4: We thank the reviewer for this valuable suggestion. In line with the comment, we have revised Figure 1 to include the distribution of Lewis Scores alongside fecal calprotectin levels, allowing for a more comprehensive visualization of the relationship between biomarker values and endoscopic findings.

To improve transparency, we have now added the minimum and maximum values for fecal calprotectin and Lewis Scores in the footnote to Table 1, complementing the medians and IQRs already presented. In addition, we have specified in the Results section the minimum and maximum values of fecal calprotectin for each inflammation subgroup, alongside the medians and IQRs (lines 183–190).

We also acknowledge the limitation related to the very small number of patients with isolated proximal disease (5%). This subgroup analysis has been reframed in the Discussion as hypothesis-generating rather than conclusive, due to the low statistical power and limited disease activity observed in these patients. Importantly, we note that this finding is consistent with prior literature, since isolated proximal Crohn’s disease represents a rare phenotype. Given its typically more aggressive clinical course, we considered it essential to report and analyze this subgroup, while clearly recognizing and discussing the limitations in interpretation (lines 267-273).

Comment 5: At line 253  The reality is that comparing a group of 4 patients with isolated proximal disease to as much as 35 with pan small bowel disease has real problems

Low statistical power: With only 4 patients, the ability to detect true differences is minimal. You might miss real effects (Type II error).

Unstable estimates: Medians and IQRs from 4 people are not reliable representations of the “population.”

Multiple comparisons penalty: Bonferroni correction makes it even harder to detect significance, so results are even less stable with such a small group.

Risk of spurious results: A single outlier among 4 patients could drive the group median and make differences look larger than they really are.

The authors need to make this very clear to the reader and discuss how this affects the conclusion that FC is less reliable in proximal disease.

Response 5: We comprehend the reviewer’s concerns regarding the very small number of patients with isolated proximal small bowel involvement (5%). As pointed out, such a small subgroup inevitably limits statistical power, increases the risk of Type II error (failure to detect real associations), and leads to unstable estimates in which medians and IQRs may not reliably represent the underlying population.

In line with this valuable feedback, we have revised the Discussion to explicitly state that the analysis of isolated proximal disease should be regarded as exploratory, with low statistical power, and that any conclusions should be interpreted with caution.

At the same time, we believe that reporting this subgroup remains clinically meaningful. Isolated proximal Crohn’s disease is a rare phenotype, which explains the small sample size across most published cohorts. In fact, even in large-scale studies such as the IBD Genetics Consortium (n = 2105), only 25 patients (1.2%) had isolated proximal disease without ileal or colonic involvement (Ref. [12] in our manuscript). Despite the limited precision of estimates, including these patients adds incremental evidence to an underexplored but clinically relevant subgroup, typically associated with a more aggressive course. For this reason, we have reframed our findings as hypothesis-generating, while emphasizing their limitations and the need for further studies in larger populations (lines 268-274 and lines 330-333).

Reviewer 2 Report

Comments and Suggestions for Authors

The authors addressed an important and a novel topic which is the role of fecal calprotectin in small bowel Crohn's disease by conducting a retrospective study. The manuscript is well written and organized. However, some issues need to be clarified.

  1. Replace the question in the title with an illustrative statement informing about the type and results of the study.
  2. Regarding the selection of patients, how did the authors confirm the diagnosis of Crohn's disease by histopathology despite using capsule endoscopy which lacks the ability to take biopsy?
  3. Regarding patients' characteristics, how did the patients receive Topical corticosteroids in small bowel Crohn's disease despite the exclusion of cases of colonic Crohn's?
  4. Did any case receive dual therapy i.e. corticosteroids and immunomodulators?
  5. What does the author mean by "Age at diagnosis" in table 1?
  6. Kindly, make a correlation between the level of fecal calprotectin and the received therapy to make the article more informative.
  7. Provide some images of the capsule endoscopy.

Author Response

The authors addressed an important and a novel topic which is the role of fecal calprotectin in small bowel Crohn's disease by conducting a retrospective study. The manuscript is well written and organized. However, some issues need to be clarified.

Comment 1: Replace the question in the title with an illustrative statement informing about the type and results of the study.

Response 1: We thank the reviewer for this valuable suggestion. In line with the recommendation, we have revised the title to remove the interrogative form and better reflect the scope and methodology of the study. The new title is: “Fecal Calprotectin and Segmental Inflammation in Small Bowel Crohn’s Disease: A Capsule Endoscopy-Based Study.”

We believe this formulation more accurately conveys the focus and design of the study, while improving clarity and readability.

Comment 2: Regarding the selection of patients, how did the authors confirm the diagnosis of Crohn's disease by histopathology despite using capsule endoscopy which lacks the ability to take biopsy?

Response 2: We thank the reviewer for raising this important point. We agree that capsule endoscopy does not allow for histological confirmation of Crohn’s disease. In our study, the diagnosis of Crohn’s disease was previously established in all included patients based on standard clinical, radiological, endoscopic, and histopathological criteria. Capsule endoscopy was performed subsequently for the assessment of small-bowel involvement and disease activity, but it was not used as a diagnostic tool to establish Crohn’s disease. We have now clarified this in the Methods section to avoid any misunderstanding (lines 74-77).

Comment 3: Regarding patients' characteristics, how did the patients receive Topical corticosteroids in small bowel Crohn's disease despite the exclusion of cases of colonic Crohn's?

Response 3: We appreciate the reviewer’s observation. In our cohort, the term “topical corticosteroids” referred to oral budesonide formulations with targeted release in the ileum and right colon, which are widely used in the management of small-bowel Crohn’s disease. We acknowledge that the term could be misleading, as it might be interpreted as rectal formulations, which were not used in any of the included patients. We have clarified this in the Table 1 footnote to ensure precision.

Comment 4: Did any case receive dual therapy i.e. corticosteroids and immunomodulators?

Response 4: We thank the reviewer for raising this point. Indeed, a small subset of patients received dual therapy. Specifically, 4 patients (4.6%) were treated with a combination of biologic therapy and immunomodulators. This information has now been explicitly added to both the Results section (patients’ characteristics – lines 161-164) and to Table 1, under the therapy category, to improve clarity. No patients were on concomitant corticosteroids and immunomodulators at the time of capsule endoscopy.

Comment 5: What does the author mean by "Age at diagnosis" in table 1?

Response 5: We thank the reviewer for pointing out this ambiguity. The variable “Age at diagnosis” in Table 1 refers to the Montreal classification subgroups:

  • A1: diagnosis before 17 years of age,
  • A2: diagnosis between 17 and 40 years,
  • A3: diagnosis after 40 years.

We have now clarified this in the Table 1 legend to avoid confusion.

Comment 6: Kindly, make a correlation between the level of fecal calprotectin and the received therapy to make the article more informative.

Response 6: We thank the reviewer for this thoughtful suggestion. Following this recommendation, we performed an exploratory analysis comparing fecal calprotectin (FC) levels across treatment groups (no therapy, corticosteroids, immunomodulators, biologics, and combined therapy). Using a Kruskal–Wallis test, no significant differences in median FC values were observed among the therapeutic groups (p = 0.316). Pairwise comparisons were also explored, but no relevant differences emerged after adjustment for multiple testing. These results have been briefly integrated into the Results section (section 3.2 – lines 206-208).

Comment 7: Provide some images of the capsule endoscopy.

Response 7: We thank the reviewer for this helpful suggestion. In line with the request, we have now added representative images from capsule endoscopy to illustrate typical inflammatory findings in the first, second, and third small bowel tertiles. These images are presented in Figure 3, providing visual examples that complement the quantitative results described in the manuscript.

Reviewer 3 Report

Comments and Suggestions for Authors

This is an interesting study examining the association between fecal calprotectin (FC) and small-bowel Crohn’s disease activity using capsule endoscopy. The focus on segment-specific inflammation is valuable, but I have several concerns that limit the strength of the conclusions:

1. Timing of FC vs SBCE
The allowance of up to 3 months between FC and capsule endoscopy is too long for a biomarker–endoscopy correlation in Crohn’s disease. Please provide the actual distribution of intervals and perform sensitivity analyses for patients tested within shorter windows (e.g., ≤30 days). This could change the interpretation, especially for the “no correlation in the first tertile” claim.

2. Small proximal subgroup
Only 4 patients had isolated proximal disease, which makes it difficult to conclude that FC is “not reliable” in this location. This should be reframed as hypothesis-generating, with confidence intervals reported.

3. Assay and handling details
Across an 8-year study period, different FC assays or protocols may have been used. Please specify the assay(s), handling, and any changes over time. If assays varied, a sensitivity analysis limited to one era would help.

4. Additional confounders
Therapy, CRP, smoking, and disease duration can all influence FC. Multivariable models (or at least adjusted analyses) would help clarify whether FC truly tracks with Lewis Score independent of these factors.

5. Clinical utility of the 178 µg/g cut-off
The ROC analysis contrasts pan-small bowel vs isolated distal disease, which may not be the most clinically actionable endpoint. Please also report performance for “any inflammation vs none” and “proximal involvement vs absent,” even if exploratory.

6. Overstatement of conclusions
The statement that FC “is not reliable” for proximal disease is too strong given the small n and potential timing/assay issues. I suggest softening this to “in this cohort, FC did not correlate with proximal inflammation.”

Author Response

Comment 1: Timing of FC vs SBCE
The allowance of up to 3 months between FC and capsule endoscopy is too long for a biomarker–endoscopy correlation in Crohn’s disease. Please provide the actual distribution of intervals and perform sensitivity analyses for patients tested within shorter windows (e.g., ≤30 days). This could change the interpretation, especially for the “no correlation in the first tertile” claim.

Response 1: We thank the reviewer for this important comment. In our cohort, the median interval between FC collection and SBCE was 51 days (IQR 23–90; range 0–90). Overall, 33 patients (37.9%) had FC collected within 30 days of SBCE, and 56 patients (64.4%) within 60 days. These data have now been added to the manuscript (lines 177–179 ).

To further address this concern, we performed sensitivity analyses restricted to patients with shorter FC–SBCE intervals. Among those tested within ≤30 days (n = 33), no significant correlation was observed between FC and LS in the 1st tertile (ρ = –0.214, p = 0.177), whereas significant correlations were confirmed in the 3rd tertile (ρ = 0.490, p = 0.004). Similarly, in the ≤60-day subgroup (n = 56), FC remained significantly correlated with LS in the 2nd (ρ = 0.363, p = 0.006) and 3rd tertiles (ρ = 0.407, p = 0.002), but not in the 1st tertile (ρ = –0.157, p = 0.248).

These findings demonstrate that the absence of correlation between FC and LS in the first tertile was consistent across sensitivity analyses and is therefore unlikely to be explained by the timing interval between FC sampling and SBCE. We have incorporated these results in the revised manuscript (Results section – correlation analyses, lines 217–222 ).

Comment 2: Small proximal subgroup
Only 4 patients had isolated proximal disease, which makes it difficult to conclude that FC is “not reliable” in this location. This should be reframed as hypothesis-generating, with confidence intervals reported.

Response 2: We thank the reviewer for this thoughtful comment. We agree that the small number of patients with isolated proximal small bowel disease (4.6%) limits the robustness of any conclusions. In line with the reviewer’s suggestion, we have reframed this finding throughout the Discussion and Conclusion sections as exploratory and hypothesis-generating.

We also acknowledge that the limited sample size in this subgroup is not unique to our study but rather reflects the rarity of isolated proximal Crohn’s disease. Even in large multicenter cohorts, this phenotype has consistently been reported as uncommon, which explains why most published studies struggle with small numbers in this subgroup. Nonetheless, because proximal involvement is often associated with a more aggressive clinical course and worse prognosis, we believe that reporting and analyzing these cases, even as exploratory data, remains of clinical importance.

Regarding confidence intervals, we appreciate the reviewer’s point and have now provided 95% confidence intervals for all ROC-derived measures (AUC, sensitivity, specificity, PPV, and NPV), to transparently reflect the uncertainty around diagnostic performance estimates. For non-parametric comparisons (Kruskal–Wallis, Mann–Whitney, Jonckheere–Terpstra), confidence intervals are not routinely generated, so results are reported as medians with IQRs, as per standard practice.

We hope that these revisions appropriately temper the strength of our conclusions, while still contributing additional data to this rare but clinically relevant phenotype.

Comment 3: Assay and handling details
Across an 8-year study period, different FC assays or protocols may have been used. Please specify the assay(s), handling, and any changes over time. If assays varied, a sensitivity analysis limited to one era would help.

Response 3: We thank the reviewer for this thoughtful comment. We confirm that all fecal calprotectin (FC) measurements in our study were performed using the Calprotectin Quantum Blue assay (Bühlmann, Schönenbuch, Switzerland), a quantitative immunochromatographic method with automated reading. Importantly, the same assay, platform, and protocol were consistently applied throughout the entire study period, with no changes in laboratory handling procedures. As such, there was no methodological heterogeneity across the 8-year timeframe, and a sensitivity analysis restricted to a specific period was not required. This clarification has now been explicitly reported in the manuscript (lines 86–89).

Comment 4: Additional confounders
Therapy, CRP, smoking, and disease duration can all influence FC. Multivariable models (or at least adjusted analyses) would help clarify whether FC truly tracks with Lewis Score independent of these factors.

Response 4: We thank the reviewer for this observation. We fully agree that therapy, CRP, smoking, and other clinical factors can influence fecal calprotectin levels, and that it is important to evaluate whether the association with the Lewis Score is independent of these potential confounders. Following this suggestion, we have now performed multivariable linear regression models including therapy, CRP, smoking status, perianal disease, age, and sex as covariates. In these models, the Lewis Score remained independently associated with fecal calprotectin levels (p = 0.036), while among the potential confounders, only CRP showed an independent association (p = 0.007). These results have been added to the manuscript (Results, lines 210–212) and discussed accordingly.

Comment 5: Clinical utility of the 178 µg/g cut-off
The ROC analysis contrasts pan-small bowel vs isolated distal disease, which may not be the most clinically actionable endpoint. Please also report performance for “any inflammation vs none” and “proximal involvement vs absent,” even if exploratory.

Response 5: We thank the reviewer for this insightful comment highlighting the importance of clinically actionable endpoints for ROC analyses. Following this suggestion, we performed additional analyses beyond the contrast of pan-small bowel versus isolated distal disease.

We have now added the diagnostic performance of fecal calprotectin for “any inflammation vs none,” “proximal involvement vs none,” and “distal involvement vs none.” These additional results have been incorporated in the Results section (lines 229–242) and discussed in the Discussion, where we acknowledge their exploratory nature given subgroup size limitations, while also emphasizing their potential clinical applicability (lines 288-298).

Comment 6: Overstatement of conclusions
The statement that FC “is not reliable” for proximal disease is too strong given the small n and potential timing/assay issues. I suggest softening this to “in this cohort, FC did not correlate with proximal inflammation.”

Response 6 : We thank the reviewer for this observation. We agree that our original wording may have conveyed stronger conclusions than warranted for the proximal subgroup, given the small sample size and inherent limitations. Following the reviewer’s thoughtful suggestion, we have revised the text to state that “in this cohort, FC did not correlate with proximal inflammation.” - lines 334-338

We believe this rephrasing more accurately reflects the exploratory nature of our findings while maintaining the clarity of the message. At the same time, we would like to highlight that addressing this rare and clinically relevant phenotype adds meaningful value to the literature, as data on isolated proximal disease remain scarce even in larger series. By softening our conclusions, as suggested, we acknowledge the limitations while ensuring that our work continues to contribute novel insights into this challenging subgroup.

Round 2

Reviewer 1 Report

Comments and Suggestions for Authors

I would like to thank the authors for their openness to addressing the small concerns I identified with the study. I hope they agree the changes they have made usefully improve the readability and applicability of the manuscript. I commend them for undertaking this interesting study and recommend it for publication to the Editor.

Reviewer 3 Report

Comments and Suggestions for Authors

thanks